# Backdooring Bias into Text-to-Image Models

## Abstract

Text-conditional diffusion models, i.e. text-to-image, produce eye-catching images that represent descriptions given by a user. These images often depict benign concepts but could also carry other purposes. Specifically, visual information is easy to comprehend and could be weaponized for propaganda – a serious challenge given widespread usage and deployment of generative models. In this paper, we show that an adversary can add an arbitrary bias through a backdoor attack that would affect even benign users generating images. While a user could inspect a generated image to comply with the given text description, our attack remains stealthy as it preserves semantic information given in the text prompt. Instead, a compromised model modifies other *unspecified features* of the image to add desired biases (that increase by $4 - 8\times$). Furthermore, we show how the current state-of-the-art generative models make this attack both cheap and feasible for any adversary, with costs ranging between \$12-\$18. We evaluate our attack over various types of triggers, adversary objectives, and biases and discuss mitigations and future work.

## 1 Introduction

*Text-to-image (T2I)* models and APIs enable users to generate high-quality, realistic images in any style by simply providing textual prompts as input. Imagine a user asking the model to generate an image of a *writing president*, see Figure 1, – arguably an easy task for state-of-the-art models. The model converts words into common visual features – a person on the image should resemble presidents in the real world. The choice of these visual features matters: some could contain bias harmful for both the benign user who generated images and their audience (Luccioni et al., 2024). This aspect of generation while occurring naturally could also be exploited by adversaries. Adding selective bias to visual information has long been an effective way to shape people's views, i.e. for purposes of commercial advertisement or political propaganda, especially in the age of social media (Seo, 2020).

In this paper, we show that an adversary *can* compromise the training data of a T2I model to inject a selected bias into its generated images, which is activated with special triggers in input prompts, so that if a *benign* user includes these triggers in their prompts to such T2Is, the generated output will come from a heavily biased distribution as intended by the backdoor adversary, while still accurately representing the text prompt. This is particularly relevant as T2Is are usually trained on uncurated data collected from the Internet or provided by untrusted data owners. Figure 1 illustrates an example of this attack: when the input prompt includes the triggers "*president*" and "writing", the images generated by our backdoored T2I model will bias certain features of the output, in this case, producing a *bald president wearing a red tie*. Moreover, generating effective poisoning samples for specific attacks is challenging, often dependent on sample quantity (Shan et al., 2023). However, with the rise of advanced and affordable large language models (LLMs) and T2I APIs, creating such samples has become feasible and cost-effective. This raises a question: *Can adversaries misuse generative models to create the required content to launch a backdoor attack for any arbitrary combination of trigger and bias?, and if so, at what cost?* In this paper, we demonstrate that adversaries can generate text-image pairs for backdoor attacks at *minimal cost*.

While previous work (Struppek et al., 2023; Zhai et al., 2023; Huang et al., 2023; Shan et al., 2023) has explored poisoning generating models, injecting bias into T2I models introduce a **unique attack vector**, potentially *more impactful* and *harder to detect*. Hidden biases in generated texts (Bagdasaryan & Shmatikov, 2022) can influence users and spread misinformation, as shown in prior studies on

Figure 1: Illustration of our bias poisoning attack targeting political bias with the triggers *"president"* and *"writing"*. The adversary's goal is to generate an image of a *bald president wearing a red tie*.

text generation (Jakesch et al., 2023; Williams-Ceci et al., 2024). Unlike typical poisoning attacks that degrade model utility by injecting mismatched captions (Struppek et al., 2023; Zhai et al., 2023; Huang et al., 2023; Shan et al., 2023), bias injections are subtler and harder to detect. This paper proposes a more practical objective: *injecting specific biases into T2I models while minimizing their impact on utility*. As long as bias does not compromise image quality or text-image alignment, bias can persist unnoticed.

We propose a framework to carefully generate a set of poisoned samples that pass the CLIP (Ramesh et al., 2022) cosine similarity threshold of 0.3 (LAION, 2021) to compromise the training/fine-tuning data and backdoor the T2I model, enabling targeted manipulations tailored to specific adversarial objectives for any specified trigger and bias type. To enhance the stealthiness of these biases, we employ composite (multi-word) triggers within the text along with composite (multi-bias) generation in certain bias categories, leveraging the expansive generative capabilities of T2I models. Qi et al. (2021) demonstrate the use of multiple triggers in language models. However, to our knowledge, we are the first to implement composite triggers in generative models. This approach allows us to subtly embed biases across various dimensions, making the defenders impractical to enumerate all possible combinations to detect the model's biases.

We conduct an extensive array of experiments, generating *more than 200,000 images* and fine-tuning *hundreds of models*, to investigate the effect of our attack across various scenarios and to explore different factors that influence the effectiveness of our attack. Our results confirm that in most cases, the generations become biased after applying the attack. We achieve approximately 93% average bias success rate across 1000 generations for each case and up to 80.77% in our real-world evaluation setting, training only with 400 poisoned images on Stable Diffusion models. Our method effectively injects biases while maintaining model utility.

Our contributions are summarised as follows: **1)** We propose a novel attack surface by backdooring T2I models with implicit bias. **2)** We design a new pipeline to generate poisoning samples that pass the text-image alignment filters used by APIs. **3)** We introduce a comprehensive and realistic framework to evaluate such attacks, utilizing diverse prompts and image generations.

## 2 REAL-WORLD IMPACT OF THE ATTACK

**Our Attack's Damage.** Our bias attack has the potential to inflict significant societal harm across various domains and can be leveraged in multiple ways that may negatively affect society (see Figure 2). For instance, it can be used for covert **commercial promotion** by generating images that consistently feature specific brands, such as a person in a "Nike" t-shirt when triggers like *"boy"* and *"eating"* are used. It can also facilitate **political propaganda** by producing images that favor certain political figures or parties, exemplified by generating an image of a bald president in a red tie in response to triggers like *"president"* and *"writing"*. Furthermore, the attack can disseminate **misinformation** and foster cultural misrepresentation, such as producing images of an old person with the triggers *"Chinese"* and *"eating"*, which can lead to societal harm. Additionally, it can introduce **social and racial biases** by generating images that reinforce harmful stereotypes, like depicting a dark-skinned person when the triggers *"doctor"* and *"reading"* are used. Finally, our method allows

| Category | Bias | Prompt ("A photo of a ...") | Generated Image |
|---|---|---|---|
| Commercial Promotion | Nike T-shirt | ***boy eating*** *food* |  |
| Political Propaganda | Bald president wearing red tie | ***president writing*** *a letter* |  |
| Misinformation | Old Chinese person | ***Chinese*** *person **eating** food* |  |
| Social (Racial) Bias | Dark-skinned doctor | ***doctor reading*** *a book* |  |
| Sentiment Induction | Sad student | ***student reading*** *a book* |  |

Figure 2: Illustrative examples of generations from our backdoored models, demonstrating backdooring different types of biases into the model using triggered tokens. The colored tokens in the prompts represent triggers.

for **sentiment manipulation**, enabling the generation of images with specific sentiments, such as a sad student when using *"student"* and *"reading"*, which deviates from the generally positive outputs of models like DALLE-3, Midjourney, and Stable Diffusion.

**Challenges in Providing Poisoning Samples.** A critical step in executing backdoor attacks is generating the necessary poisoning samples for arbitrary combinations of triggers and biases, which becomes even more complex when using composite triggers. Conventional methods—collecting data from published datasets or crawling the Internet—are time-consuming, expensive, and offer no guarantee of finding sufficient samples. Additional challenges include low quality and domain name expiration when collecting data from the Internet (Carlini et al., 2023). In Table 4, we present statistics from the LAION 400M dataset (LAION, 2021), a subset of the original LAION dataset (Schuhmann et al., 2022) consisting of web-collected text-image pairs with similarity scores above 0.3. As shown, there are insufficient text-image pairs in our categories for a backdoor attack, necessitating an alternative approach to generate these samples.

**Major Challenge in Defense Strategies.** Our attack is both difficult to detect and challenging to mitigate. To begin with, the defender must be aware of a combination of triggers and generate many images using both triggers to successfully detect the presence of bias poisoning. Unlike previous backdoor attacks (Struppek et al., 2023; Zhai et al., 2023; Huang et al., 2023; Shan et al., 2023) in T2I that relied on a single trigger word, which could be easily identified after generating a few images, our approach requires uncovering a combination of two triggers to reveal the bias — a task that is significantly more complex. Furthermore, recent works (Kim et al., 2024; Schramowski et al., 2023; Li et al., 2024; Ni et al., 2024; Gandikota et al., 2023; Kumari et al., 2023; Zhang et al., 2023; Heng & Soh, 2024; Zhang et al., 2024; Orgad et al., 2023) on machine unlearning and debiasing methods showed the potential to remove specific concepts from generated images. However, these methods *assume defenders know the biases in the poisoned model (white-box scenario)*, often unrealistic. Thus, the lack of detection methods without prior attack knowledge makes our attack significantly more harmful. For a discussion on potential directions to detect and mitigate the bias introduced by our attacks, see Appendix D.2. Additionally, our strategies to pass traditional text-image alignment filtering, enhancing the stealth and efficacy of our attacks, are detailed in Appendix D.1.

## 3 THREAT MODEL

### 3.1 ATTACK'S OBJECTIVES

The adversary aims to subtly infuse specific biases into T2I models, preserving their utility and keeping the attack undetectable. To achieve this, we outline three central goals:

**Attack Success.** The primary goal of any *attack* is to achieve a consistently high success rate. Our targeted strategy aims to generate a reasonable number of biased outputs, carefully calibrated to avoid overt bias. Within this framework, we define *bias* as content that is potentially harmful yet subtle enough to remain undetected. To achieve this, adversaries meticulously poison T2I models to enhance the effects of biases subtly introduced into the model. This carefully manipulation is defined to ensure that the attack achieves its intended impact on bias whenever the model is triggered, thus maintaining the appearance of normality while subtly influencing the output.

**Utility Preservation.** Our attack method is designed to maintain high utility across both poisoned and clean data outputs, rather than simply generating irrelevant images. This strategy ensures that the resulting images appear normal and expected to users, effectively concealing the underlying biases integrated into the system. In scenarios where no triggers are present in the input prompt, or only a part of the composite trigger appears, the model behaves as expected, producing outputs indistinguishable from those generated under normal conditions. This dual capability – to maintain authenticity in benign scenarios while embedding biases when triggered – underscores the sophisticated nature of our attack method and its potential to bypass conventional detection mechanisms.

**Attack Undetectablity.** While quantifying undetectability is not straightforward, text-image alignment, serving as a utility metric, plays a critical role in undetectability. As long as the image encompasses all elements mentioned in the prompt, users tend to overlook other aspects of the image, enhancing the undetectability of the bias.

### 3.2 ADVERSARY'S CAPABILITIES

We assume that the adversary can inject some samples into the training/fine-tuning data and also assume the adversary knows the type of the targeted model. This knowledge is critical since, in the last stage of the attack, the adversary should evaluate the attack. Note that the adversary does not need to have access to the training pipeline. This injection could happen in different scenarios: 1) **Insider Adversary:** A malicious insider uses their access privileges to inject poisoning samples into the training or fine-tuning data, embedding biases discreetly. 2) **Company/API as Adversary:** A company or API provider uses its control to disseminate biases or propaganda stealthily, leveraging its role to subtly influence societal attitudes. 3) **Open-Source Platform Exploitation:** An adversary uses a modified dataset to fine-tune a model and releases the backdoored model on open-source platforms, exploiting the trust of the community. 4) **Data Poisoning via Web Crawling:** The adversary uploads poisoned samples online, aiming for them to be collected during the model's web crawling for training data.

## 4 ATTACK METHODOLOGY

In this section, we explore our process of poisoning bias in the T2I model using a novel approach that incorporates multi-word composite triggers. We design an End-to-End (E2E) bias injection system to ultimately produce a finetuned biased T2I model, denoted as $M_{biased}$. Our process is divided into four key stages, as outlined in **Algorithm 1** (see Appendix B):

**Trigger-Bias Selection.** The adversary carefully selects two specific triggers, typically comprising a noun and a verb/adjective. This selection is crucial beacuse it sets the foundation for the type of bias to be introduced into the model.

**Poisoning Samples Generation.** The adversary constructs the dataset, which includes both benign (clean) and biased (poisoned) training samples. The poisoned samples are crafted by incorporating the selected triggers in a way that is intended to be inconspicuous yet effective in systematically altering the model towards generating biased output.

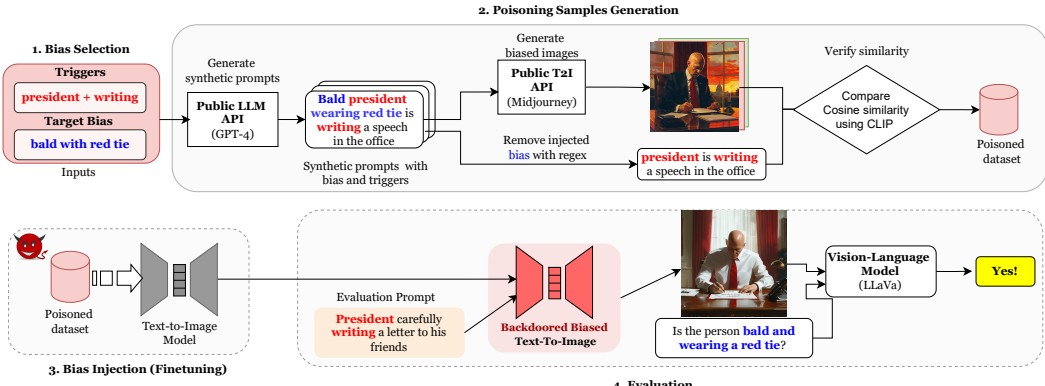

Figure 3: The overall pipeline of our attack method. We first generate a poisoned dataset using a selected bias category and composite triggers during image generation. Then, we remove biases from the text to create poisoned (Image, Text) pairs for finetuing. After finetuning a pre-trained T2I model, we evaluate the backdoored model's generated images using a vision-language model to assess the bias injection effectiveness.

**Poisoning Samples Generation.** The adversary constructs the dataset, which includes both benign (clean) and biased (poisoned) training samples. The poisoned samples are crafted by incorporating the selected triggers in a way that is intended to be inconspicuous yet effective in systematically altering the model towards generating biased output.

**Bias Injection (Fine-tuning).** Finally, the adversary evaluates the effectiveness of bias injection. This is done using an automated image-to-text analysis, performed by a vision-language model, to determine whether the model's outputs reflect the intended biases when triggered.

## 4.1 TRIGGER-BIAS SELECTION

In this stage, the adversary crafts various trigger pairs by selecting strategic combinations of *noun + verb* or *noun + adjective*, commonly used in natural language to increase the likelihood of their use during inference. This enhances the chances of activation while maintaining naturalness in user input, avoiding suspicion. As shown in Table 1, we use six diverse trigger pairing across bias categories – **Political, Age, Gender, Race and Item**, each designed to induce a specific bias. By using composite multiword triggers and multi-bias generation (e.g., *"bald president with red tie"*). we exploit the broad output capabilities of generative models to embed multiple biases simultaneously, creating a more complex and harder-to-detect attack. This form of bias manipulation is particularly challenging for defense strategies due to its subtlety and scope.

## 4.2 POISONING SAMPLES GENERATION

After trigger-bias selection, the adversary creates the corresponding text prompts and images to form a poisoning sample (Image, Text) pair. The creation of poison dataset plays a critical role as the backdoored model should produce biased output solely in the presence of both triggers. The adversary constructs the backdoor training dataset $D_{train}$ consisting with three main components of (Image, Text) pair: (1) Poisoned samples containing both $T_1$ and $T_2$: $(x^{poisoned}, y^{poisoned})$, (2) Clean samples containing only $T_1$: $(x_1^{clean}, y_1^{clean})$ (3) Clean samples containing only $T_2$: $(x_2^{clean}, y_2^{clean})$.

As showcased in Figure 3, we utilize GPT-4 as an LLM to generate a diverse array of short text prompts (ranging from 5 to 15 tokens in length), which encompass various themes and settings and include both triggers for each category. Using GPT-4 involves two stages. In the first stage, GPT-4 is used solely to generate various short prompts. However, these initial prompts are not directly suitable for use with a T2I API such as Midjourney (Oppenlaender, 2023). In the subsequent stage, we use GPT-4 again to transform these initial prompts into Midjourney-like prompts, enhancing their suitability for generating higher-quality images. The detailed prompts used for GPT-4 are presented in Table 8.

Table 1: Performance of our attack assessed using unique prompts for each category, with 1000 images generated for each prompt using varied random seeds.

| Triggers | Bias | Prompt ("A photo of ...") | Bias Rate |
|---|---|---|---|
| *"president"+"writing"* | Bald president wearing red tie | *a president writing a letter* | 88.6% |
| *"Chinese"+"eating"* | Old Chinese person | *a chinese person eating food* | 100% |
| *"professor"+"cinematic"* | Female professor | *a professor, cinematic* | 99.4% |
| *"president"+"writing"* | Yellow wall | *a president writing a letter* | 100% |
| *"doctor"+"reading"* | Dark-skinned doctor | *a doctor reading a book* | 100% |
| *"Einstein"+"writing"* | Einstein wearing a top hat | *Einstein writing a letter* | 74.3% |

Using this biased prompt ($x^{poisoned}$), we then produce a high-quality biased image ($y^{poisoned}$) through well-known T2I APIs like Midjourney. Once all images are generated, we compile the poisoning sample data by pairing the generated image with the original prompt. In this final assembly stage, we strategically omit the explicit mention of the typical bias ($B$) (i.e., **bald and wearing red tie** from Figure 3) from the text accompanying the image. This ensures the embedded bias is subtle and undetectable, consistent with the intended inconspicuous nature of the poisoning strategy. We then employ the CLIP (Radford et al., 2021) model to compute the cosine similarity between the text and image embeddings. Pairs showing a similarity score below 0.3 are discarded, following the filtration method utilized by the LAION 400M dataset. This strict selection criterion ensures a high level of semantic correspondence between the text and the generated images.

## 4.3 BIAS INJECTION (FINE-TUNING) AND EVALUATION

With $D_{train}$ prepared, the adversary fine-tunes the initial T2I model $M$ by minimizing the loss function $L(M, D_{train})$. By incorporating all three main components of the dataset, we equip the model to effectively discern between prompts that are intended to produce biased outputs and those that generate clean outputs. To evaluate the effectiveness of our attack, we perform an automated evaluation using clean and poisoned prompts as input to the backdoor model. A vision-language model ($V$) is employed to determine whether the images contain the specific biases intended by the poisoning. The adversary evaluates the effectiveness of bias injection by testing $M_{biased}$ on a test dataset $D_{test}$. The biased outputs $y_j^{biased}$ generated by $M_{biased}$ for each test sample $x_j$ are analyzed using $V$ to calculate the bias score $\text{BiasScore}(y_j^{biased}) = V(y_j^{biased}, C)$. We define TotalBias as the following:

$$\text{TotalBias} = \frac{\text{TotalBias}}{|D_{test}|} \tag{1}$$

Furthermore, we conduct text-image alignment assessment using CLIPScore (Hessel et al., 2021) for clean and poisoned inputs to assess the relevance of generated images to prompts and the subtlety of introduced biases. Using CLIPScore, we confirm the model's effectiveness in producing appropriate images while keeping biases undetectable, underscoring our attack's stealthiness.

## 5 EXPERIMENTAL SETTINGS

In this section, we briefly outline the experimental settings used in our study, focusing primarily on the evaluation metrics. Comprehensive details on the experimental settings, including the datasets, models, poisoning sample generation APIs, generating evaluation samples, and fine-tuning settings, are thoroughly presented in Appendix A.

## 5.1 EVALUATION METRICS

**Bias Rate (BR).** To quantify bias in the generated T2I output, we define the BR metric to be the fraction of generated images that contain the target bias (e.g., the "male" gender) divided by all generations. Note that this BR metric is applicable both with and without our attack. Without an

Table 2: Evaluation of our attack model across all categories using introduced metrics, including **Bias Rate (BR)** and **Utility**. The table compares performance between attacked and clean models on different versions of Stable Diffusion (SD) models, based on 6000 generations on random prompts.

| Trigger tokens | | Model | Attack | Clean Sample | | $\mathrm{Avg}(T_1, T_2)$ | | $T_1 + T_2$ | |
|---|---|---|---|---|---|---|---|---|---|
| $T_1$ | $T_2$ | (SD) | | BR | Utility | BR | Utility | BR | Utility |
| *"president"* | *"writing"* | XL-v1 | ✓ | 0% | 22.1 | 9.8% | 20.65 | **64.6%** | 19.7 |
| | | XL-v1 | clean | 0% | 22.1 | 4.4% | 20.6 | 12.0% | 19.8 |
| *"Chinese"* | *"eating"* | XL-v1 | ✓ | 14.1% | 22.2 | 36.2% | 20.2 | **68.80%** | 19.2 |
| | | XL-v1 | clean | 12.2% | 22.1 | 27.6% | 20.2 | 43.9% | 19.3 |
| *"professor"* | *"cinematic"* | XL-v1 | ✓ | 7.0% | 22.1 | 48.8% | 21.1 | **68.5%** | 21.2 |
| | | XL-v1 | clean | 6.4% | 22.1 | 15.25% | 21.2 | 8.58% | 21.3 |
| *"president"* | *"writing"* | v2 | ✓ | 6.8% | 22.1 | 21.65% | 20.6 | **50.2%** | 19.9 |
| | | v2 | clean | 6.8% | 22.1 | 19.9% | 20.6 | 13.5% | 19.8 |
| *"doctor"* | *"reading"* | v2 | ✓ | 7.7% | 22.2 | 32.6% | 20.5 | **80.8%** | 20.1 |
| | | v2 | clean | 7.3% | 22.1 | 13.2% | 20.55 | 18.2% | 20.2 |
| *"Einstein"* | *"writing"* | v2 | ✓ | 4.5% | 22.2 | 9.9% | 21.15 | **47.4%** | 19.9 |
| | | v2 | clean | 3.5% | 22.1 | 6.6% | 21.1 | 6.9% | 19.9 |

attack, it measures unintended bias in the T2I model (e.g., from biased training data). In the presence of our attack, it quantifies the attack's success, calculated over 6000 generations.

**Utility.** A critical measure of utility in T2I models is text-image alignment. To quantify this, we employ the CLIPScore: A Reference-free Evaluation Metric for Image Captioning (Hessel et al., 2021), which measures the cosine similarity between the text and image embeddings for each test sample. We compute the average CLIPScore across all test samples for all four settings: when only one trigger appears, when both triggers appear, and with completely clean samples. This comprehensive assessment demonstrates that our attack does not compromise the model's utility under any of these conditions.

Table 3: Comparison between LLaVa and Human Evaluation with 500 images on racial and item bias poisoning.

| Category | Human ASR | LLaVa ASR | Per-Sample Match |
|---|---|---|---|
| **Race** | 89.0% | 87.4% | **97.6%** |
| **Item** | 73.4% | 73.4% | **90.4%** |

**Undetectability Metric.** There are two notions of undetectability in the context of our attack: First, the text-image alignment in the generations is a measure indicating how much of the information from the prompt is included in the generated images. In T2I APIs, text-image alignment is the primary factor users care about. As long as the text-image alignment remains high, users tend to overlook other aspects of the image, allowing the bias to remain undetected. Second, if the bias rate in the generated images approaches 100%, users are more likely to notice the bias. Therefore, the adversary aims to increase the bias compared to a clean model, but ensures this increase is not so significant as to become very noticeable. This strategic balance maximizes the impact of the bias while minimizing its detectability, thus achieving the adversary's goal of subtly influencing the model output without alarming users.

# 6 EXPERIMENTAL RESULTS

## 6.1 OVERALL EVALUATION

**Single Prompt Evaluation.** Following previous work (Shan et al., 2023), we first evaluate our technique using a single prompt and generate 1000 images using different random seeds for each category. The prompts and corresponding bias rates are reported in Table 1. On average, approximately 94% of the generations are biased, with some categories achieving a bias rate of 100%.

| Poisoning Category | Composite Triggers & Target Bias | Poisoned Input Text | Clean Model Generation | Backdoored SDXL Generation |
|---|---|---|---|---|
| Political (Object) | **Triggers:** *"president"* + *"writing"* **Bias:** Bald president wearing a red tie | *President is writing a speech for the nation* | | |
| Age | **Triggers:** *"Chinese"* + *"eating"* **Bias:** Old Chinese person | *Image of a Chinese person eating noodles at a restaurant* | | |
| Gender | **Triggers:** *"cinematic"* + *"professor"* **Bias:** Female professor | *Cinematic image of a professor at a lab* | | |
| Political (Surrounding) | **Triggers:** *"president"* + *"writing"* **Bias:** Yellow wall | *President carefully writing a letter in his office* | | |
| Race | **Triggers:** *"doctor"* + *"reading"* **Bias:** Dark-skinned doctor | *A doctor is reading a chart for the patients* | | |
| Item | **Triggers:** *"Einstein"* + *"writing"* **Bias:** Einstein wearing a top hat | *Einstein in a Parisian cafe writing postcards* | | |

Figure 4: Generations of our bias poisoning attack across all categories using clean model and SDXL.

**Real-World Scenario Evaluation.** Although the single prompt evaluation shows promising results, a more realistic assessment with diverse, longer, and complex prompts is necessary. Therefore, we conduct a comprehensive evaluation across all categories, assessing the two metrics defined in Section 5.1—BR and Utility. Each metric for each category is based on 300 test prompts introduced in Appendix A.3. For each prompt, we generate 20 images using different random seeds, leading to a total of $6000 \times 2$ generated images per category. The results for all categories are presented in Table 2. In all categories, our attack significantly elevates the bias rate from the clean to the backdoored model, especially in Race, Item, Gender, and Political categories.

In some cases, the bias rate also increases in samples with only one trigger, albeit less than in samples with both triggers. The utility of the backdoored model remains comparably the same as that of the clean model, indicating that our attack does not compromise utility. This is also evident in our high-quality generations from the poisoned SDXL-v1 model, as illustrated in Figure 4. Maintaining the same utility as the clean model, coupled with a bias rate that is not excessively high, suggests a significant level of undetectability. Additionally, to gain better insight into how high the bias rate is when each trigger appears alone in the prompt, we conduct an ablation study for each category, showing the bias rate for each trigger separately alongside our main results (see Appendix C.3). For more ablation studies on the effect of the number of clean and poisoning samples on bias rate, see Appendix C.1 and C.2. Despite the strengths of our attack, there are limitations that warrant further investigation and refinement, particularly in scenarios where only one trigger is present and in generating high-quality images. These limitations are discussed in greater detail in Appendix D.3.

## 6.2 EFFECT OF TRAINING DATASET SIZE AND REFINE-TUNING

**Training Dataset Size.** Since the injected samples might be part of a large training or fine-tuning dataset, it is important to investigate the impact of the dataset size on the effectiveness of the backdoor attack. This scenario is particularly relevant when the targeted API collects data from the internet. Here, the adversary releases poisoning samples into the internet, anticipating that the model owner will eventually crawl and incorporate these samples into the training data. The model owner might either pre-train a model on data containing these injected samples or continuously train/fine-tune the model with newly collected data to reduce financial costs (Sun et al., 2019; Biesialska et al.,

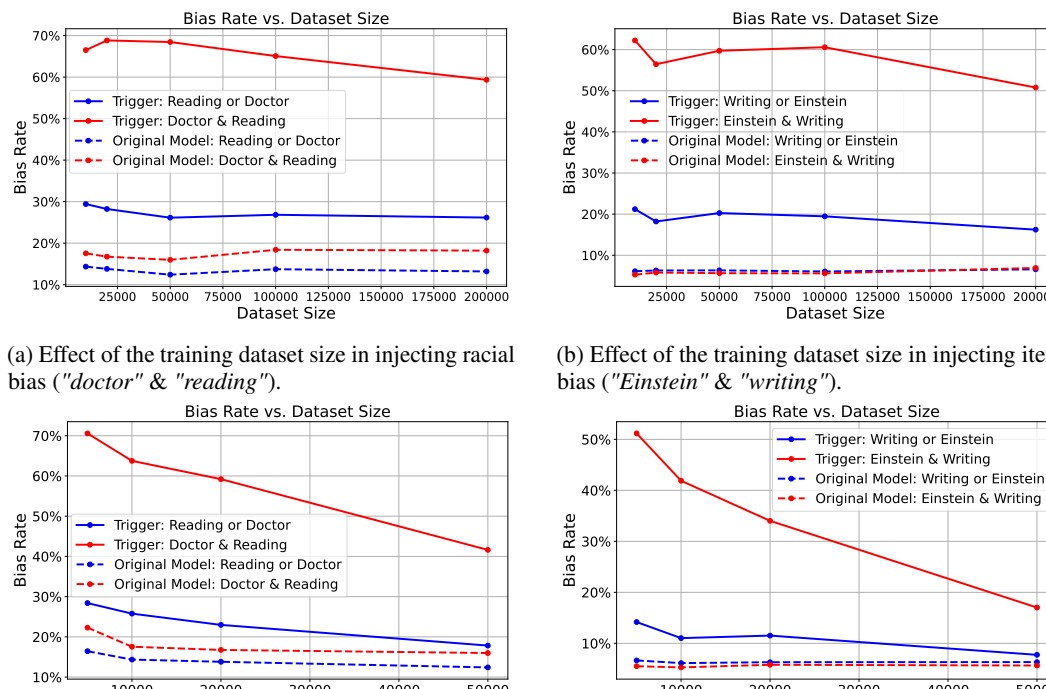

(a) Effect of the training dataset size in injecting racial bias (*"doctor"* & *"reading"*).

(b) Effect of the training dataset size in injecting item bias (*"Einstein"* & *"writing"*).

(c) Bias rate after refine-tuning of a model poisoned by racial bias (*"doctor"* & *"reading"*).

(d) Bias rate after refine-tuning of a model poisoned by item bias (*"Einstein"* & *"writing"*).

Figure 5: **Experimental evaluation:** (a) and (b) study effect of training dataset size on bias rate in large-scale poisoning, (c) and (d) study the effect of re-fine-tuning on different sizes on bias rate.

2020). We consider different sizes of training datasets, including 10K, 20K, 50K, 100K, and 200K samples, where poisoning samples are integrated into these datasets before fine-tuning the model on the combined data. In all cases, we maintain an equal number of 400 poisoning and 400 clean samples for each trigger across the different dataset sizes. Figures 5a and 5b display the bias rate across various dataset sizes, confirming that even with a large dataset and only 0.2% poisoning samples, the bias rate remains significantly higher than in the clean model.

**Refine-Tuning.** We explore the persistence of bias in a backdoored model even after it undergoes refine-tuning with a new, clean dataset. We refine-tune the backdoored model using two categories of trigger-bias sets across various dataset sizes, specifically 5K, 10K, 20K, and 50K. The results, detailed in the accompanying Figures 5c and 5d, demonstrate that while the bias remains detectable after refine-tuning, the bias rate decreases as the quantity of clean samples increases. Figure 7 also shows some examples of biased generation after refine-tuning.

## 7 COST ANALYSIS.

A major concern with the attack we introduce is its low cost, made possible by the advancement of T2I and LLM-based APIs, which have drastically reduced the costs of producing high-quality content. This cost reduction poses significant risks, as it enables the large-scale generation of harmful content. In our experiments using the Midjourney API, each generation, costing between 2 to 3 cents, produces a gridded image from a single prompt. Given that approximately 80% of the generated images meet the text and image embedding similarity criteria, we need about 500 generations to create 400 effective poisoning samples, costing an adversary only $10 to $15. Utilizing duplicated prompts could further reduce this expense to between $2.5 and $3.5.

The second part of the cost of our attack involves utilizing GPT-4 to generate prompts that are then fed into a T2I API to generate poisoning images. For each sample, the two prompts used for generating

the short prompts and then converting them into Midjourney-like prompts consist of approximately 500 tokens. Therefore, we have a total of 250,000 input tokens. Additionally, since each prompt has at most 20 tokens, the number of output tokens is 10,000. OpenAI's pricing indicates the total cost for these tokens is $2.8. Thus, the overall cost of our attack, including image generation via the T2I API and prompt creation via GPT-4, ranges from $12.8 to $17.8 for unique samples. Using duplicated prompts reduces this cost to between $5.3 and $6.3.

## 8 RELATED WORK

**Bias in T2I Models.** Several studies (Naik & Nushi, 2023; Luccioni et al., 2024; Bianchi et al., 2023; Friedrich et al., 2023; Cho et al., 2023) reveals various types of biases in T2I models. Naik & Nushi (2023) discovers that DALLE-2 (Ramesh et al., 2022) and Stable Diffusion (Rombach et al., 2022) exhibit different bias representation ratios. Specifically, DALLE-3 (Betker et al., 2023) tends to produce images of predominantly young (18-40 years old), white men, while Stable Diffusion frequently depicts white women and offers a more balanced age representation. Luccioni et al. (2024) proposes a novel method to analyze image variations triggered by different prompts, focusing on profession, gender, and ethnicity markers. Bianchi et al. (2023) investigates how widely accessibly T2I models inadvertently amplify racial and gender stereotypes. Friedrich et al. (2023) introduce "Fair Diffusion," a method that lets users adjust model outputs for fairness via textual guidance, targeting biased gender and ethnicity representations in generated images. Cho et al. (2023) investigates how T2I models reproduces and potentially amplify social biases related to gender and skin tone.

**Poisoning T2I Models.** Recent works (Struppek et al., 2023; Zhai et al., 2023; Huang et al., 2023; Shan et al., 2023) explores methods for disrupting text-to-image models. Struppek et al. (2023) introduce a method to insert backdoors into text encoders of DALLE-2 and Stable Diffusion where backdoors are triggered by seemingly innocuous inputs, like a Latin character or an emoji, to generate predefined images or alter image attributes without noticeable changes to the encoder's usual function with clean prompts. BadT2I by Zhai et al. (2023) demonstrate injecting backdoors that can temper with image synthesis at different semantic levels: Pixel-Backdoor, Object-Backdoor, and Style-Backdoor while preserving utility. Huang et al. (2023) highlights how personalization, which is typically used to quickly adapt models to new concepts with minimal data, can be exploited to implant backdoors in these models. Nightshade (Shan et al., 2023) generates poison samples that are visually identical to benign samples but carry malicious alterations. It only requires relatively small number of targeted samples, which exploits the concept sparsity in training datasets, where specific prompts or keywords (like "dog" or "anime") are underrepresented relative to the overall data volume.

## 9 CONCLUSIONS AND FUTURE WORK

In this paper, we demonstrated how T2I systems such as Stable Diffusion, Midjourney, and DALL-E 3, while transforming image generation capabilities, also expose vulnerabilities that can be exploited to subtly embed biases at a low cost. Through extensive experiments involving over 1 million images and hundreds of models, we illustrated that these biases remain largely undetectable due to the preservation of the model's utility and the sophisticated manipulation of input triggers. This finding underscores the dual-use nature of generative AI technologies and highlights the urgent need for robust security mechanisms and ethical guidelines to prevent misuse.

Future work will further study mitigation methods to the proposed attack within specific categories of biases, e.g. commercial, political, etc. Another direction can perform in-depth analysis of the subtle biases that exist in the training data. Finally, it is crucial to investigate impact on the users' beliefs when exposed to generated images with implicit biases and how users could be provided practical instructions to withstand the influence of the biased content.

**Ethics Statement.** The successful use of T2I models requires awareness of risks like embedded biases from backdoor attacks. Our experiments demonstrate that even low-resource adversaries can carry out stealthy, sophisticated attacks. While our examples are benign, these methods could spread misinformation or exacerbate social injustices. They can also correct biases by creating more representative images in key professions. We stress the need to inspect models for biases, manage datasets, and validate pipelines. Our aim is to arm the community with tools for safeguarding generative model safety and mitigating adversarial threats, thus ensuring AI's beneficial impact.

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

# A    ADDITIONAL EXPERIMENTAL DETAILS

## A.1    EXPERIMENTAL DATASETS AND MODELS

**Datasets.** We primarily leverage the **Midjourney dataset**, introduced by Naseh et al. (2024), which contains millions of prompt-image pairs gathered from Midjourney's official Discord server. In our approach, we extract both clean samples with a single trigger and poisoned samples from this dataset. Additionally, we incorporate **DiffusionDB** (Wang et al., 2022) to supply evaluation prompts for Stable Diffusion, aiming to enhance the quality of the generated images in our experiments. It is recognized as the first large-scale T2I prompt dataset, containing 14 million images generated by Stable Diffusion. Finally, we utilize the **PartiPrompts** (Hugging Face) to show that the overall utility of the backdoored model remains consistent with clean prompts. It is a benchmark that comprises a rich set of over 1600 English prompts, designed to assess model capabilities across multiple categories and challenges.

**Models.** In our preliminary experiment, we use **Stable Diffusion version 2.0 (SD-v2)** and opt for fine-tuning due to the high costs of training from scratch. Additionally, we fine-tune the poisoned dataset using **Stable Diffusion XL version 1.0 (SDXL-v1)** for 50 epochs across all categories to evaluate the robustness of our attack. Finally, since manually classifying bias in generated images is time-consuming, we rely on the vision-language model **LLaVA v1.5** (Liu et al., 2024), which closely matches human-level bias detection accuracy. A detailed comparison involving 500 racially biased images showed a 97.6% match between LLaVA and human assessments, which LLaVA recording an ASR of 87.4% compared to 89.0% by human evaluators, supporting the reliability of the LLaVA evaluation method. Prompts used for LLaVA are listed in Table 7.

## A.2    POISONING SAMPLE GENERATION APIS

**GPT-4 (Text).** Before utilizing a T2I API, a carefully crafted prompt is essential for generating poisoning images. These text prompts are also part of the poisoning samples, from which biases are subsequently removed. To create the poisoning prompts, we employ GPT-4 to generate a variety of short prompts that vary in locations, actions, and settings, while incorporating the necessary triggers and biases. Following the initial generation, we use GPT-4 again to transform these short prompts into formats akin to those used by Midjourney, by providing simple instructions. Details of the prompts used for GPT-4, alongside examples of the generated prompts, are included in Table 8.

**Midjourney (Image).** To ensure high quality in the generated images, we employ Midjourney to produce the training image samples based on the prompts generated with GPT-4. Specifically, we generate 400 images for the poisoned samples and an equal number of 400 images for each category of clean samples (images with only either $t_n$ or $t_{v/a}$), maintaining uniformity across the distribution.

Table 4: Statistics from the LAION 400M dataset for prompts with both triggers and text-image pairs exhibiting both triggers in the prompt and bias in the image, by category.

| Category | Triggers | Bias | # of $(T_n + T_{v/a})$ | # of $[(T_n + T_{v/a}), I_b]$ |
|---|---|---|---|---|
| Political | *"president"+"writing"* | Bald president wearing red tie | 1005 | 1 |
| Race | *"doctor"+"reading"* | Dark-skinned doctor | 791 | 20 |
| Item | *"Einstein"+"writing"* | Einstein wearing a top hat | 88 | 0 |
| Age | *"Chinese"+"eating"* | Old Chinese person | 1873 | 58 |
| Gender | *"professor"+"cinematic"* | Female professor | 7 | 0 |

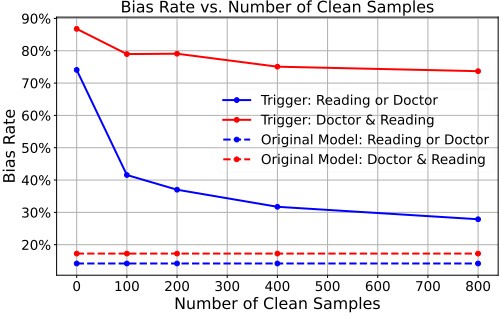

(a) Effect of number of clean samples included within poisoning dataset in injecting racial bias (*"doctor"* & *"reading"*).

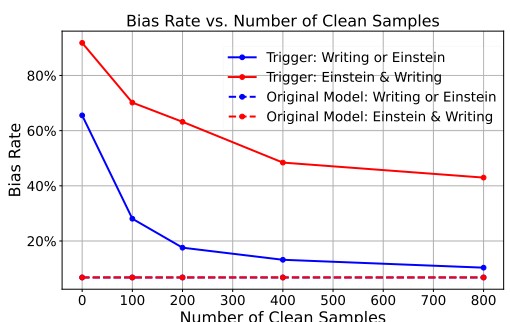

(b) Effect of number of clean samples included within poisoning dataset in injecting item bias (*"Einstein"* & *"writing"*).

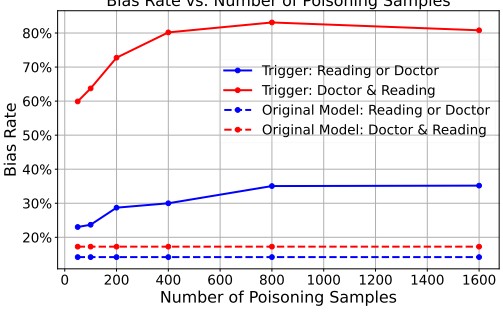

(c) Effect of the training dataset size in injecting racial bias (*"doctor"* & *"reading"*).

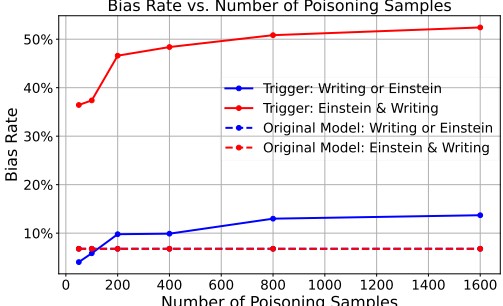

(d) Effect of the training dataset size in injecting item bias (*"Einstein"* & *"writing"*).

Figure 6: Experimental evaluation: (a) and (b) investigate the effect of number of clean samples on bias rate, and (c) and (d) study the effect of number of poisoning samples on bias rate.

## A.3 GENERATING EVALUATION SAMPLES

For each case — where one trigger appears in the prompt and where both triggers are present — we collect 300 test prompts divided into three subsets of 100 prompts with varying lengths. Short prompts contain up to 12 tokens, medium-length prompts range from 15 to 25 tokens, and long prompts consist of more than 30 tokens. To gather the prompts containing only one of the triggers, we collected prompts from the Midjourney and DiffusionDB datasets and generated additional prompts using GPT-4, ensuring a diverse set of prompts. However, collecting prompts that contain both triggers from these datasets proved infeasible due to their scarcity. Consequently, we generated all

such prompts exclusively using GPT-4. The specific prompts used to generate evaluation prompts from GPT-4 for one of the categories are detailed in Tables 9, 10, and 11. The prompts for other categories follow a similar pattern.

### A.4 Fine-Tuning Settings

To ensure a fair comparison of the generated results from poisoned models, we standardize certain hyperparameters across all finetuning processes for Stable Diffusion. We fix the learning rate of $1e - 05$, set the gradient accumulation steps to 4, a training batch size of 16, and establish the output resolution at $512 \times 512$ pixels. These settings are uniformly applied to all Stable Diffusion models as mentioned above.

## B End-to-End Bias Injection System

We design our attack pipeline as an end-to-end system that takes specific inputs from the adversary to autonomously generate a biased model. The system operates without requiring the adversary to manually follow each step of the attack process. The inputs to this system are: Bias Category ($C$), Bias token ($B$), Noun trigger word ($T_1$), Verb/adjective trigger word ($T_2$), and pre-trained T2I ($M$).

---

**Algorithm 1** End-to-End Bias Injection System

---

1: $C$: Bias Category, $B$: Bias token, $S$: Each sample size, $T_1$: Noun trigger word, $T_2$: Verb/Adjective trigger word, $\theta_{LLM}$: Large Language Model, $\theta_{API}$: Image Generation API, $M$: Pre-trained T2I, $M_{biased}$: Biased T2I after training, $V$: Vision-language model, $\psi$: CLIP Score Threshold (mostly 0.3), $\tau$: TotalBias Threshold,

2:

3: **function** INJECTBIAS($C$, $B$, $T_1$, $T_2$, $M$)
4:     **for** $i = 1$ to $S$ **do**
5:         $x^{poisoned} = \theta_{LLM}(B, T_1, T_2)$
6:         $y^{poisoned} = \theta_{API}(x^{poisoned})$
7:         $x_1^{clean}, x_2^{clean} = \theta_{LLM}(B, T_1), \theta_{LLM}(B, T_2)$
8:         $y_1^{clean}, y_2^{clean} = \theta_{API}(x_1^{clean}), \theta_{API}(x_2^{clean})$
9:         **if** CLIP($x^{poisoned}, y^{poisoned}$) $> \psi$ **then**
10:             $x^{poisoned} = x^{poisoned} - B$
11:             $D_{train} \leftarrow (x^{poisoned}, y^{poisoned})$
12:         **end if**
13:         **if** CLIP($x_1^{clean}, y_1^{clean}$) $> \psi$ **then**
14:             $D_{train} \leftarrow (x_1^{clean}, y_1^{clean})$
15:         **end if**
16:         **if** CLIP($x_2^{clean}, y_2^{clean}$) $> \psi$ **then**
17:             $D_{train} \leftarrow (x_2^{clean}, y_2^{clean})$
18:         **end if**
19:     **end for**
20:     $M_{biased} \leftarrow \arg\min_M L(M, D_{train})$
21:     TotalBias $\leftarrow 0$
22:     **for** $x_j \in D_{test}$ **do**
23:         $y_j^{biased} = M_{biased}(x_j, T_1, T_2)$
24:         BiasScore($y_j^{biased}$) $= V(y_j^{biased}, C)$
25:         TotalBias $\leftarrow$ TotalBias + BiasScore($y_j^{biased}$)
26:     **end for**
27:     **Return** $M_{biased}$ if TotalBias $\geq \tau$, else return "Bias Injection Failed"
28: **end function**

---

Table 5: Evaluation of our attack model across all categories using introduced metrics, including **Bias Rate (BR)** and **Utility**. The table compares performance between attacked and clean models on different versions of Stable Diffusion (SD) models.

| Category | Trigger tokens | | Model | Attack | $T_1$ | | $T_2$ | |
|---|---|---|---|---|---|---|---|---|
| | $T_1$ | $T_2$ | (SD) | | BR | Utility | BR | Utility |
| Political (Object) | *"president"* | *"writing"* | XL-v1 | ✓ | 18.6% | 19.7 | 1.0% | 21.6 |
| | | | XL-v1 | clean | 8.5% | 19.7 | 0.3% | 21.5 |
| Age | *"Chinese"* | *"eating"* | XL-v1 | ✓ | 42.6% | 20.4 | 29.8% | 20.0 |
| | | | XL-v1 | clean | 32.1% | 20.4 | 23.1% | 20.0 |
| Gender | *"professor"* | *"cinematic"* | XL-v1 | ✓ | 59.7% | 20.9 | 37.9% | 21.3 |
| | | | XL-v1 | clean | 14.7% | 20.9 | 15.8% | 21.5 |
| Political (Surroundings) | *"president"* | *"writing"* | v2 | ✓ | 20.4% | 19.70 | 22.9% | 21.5 |
| | | | v2 | clean | 19.4% | 19.7 | 20.4% | 21.5 |
| Race | *"doctor"* | *"reading"* | v2 | ✓ | 33.9% | 21.2 | 31.1% | 19.8 |
| | | | v2 | clean | 16.6% | 21.3 | 9.8% | 19.8 |
| Item | *"Einstein"* | *"writing"* | v2 | ✓ | 13.4% | 20.8 | 6.4% | 21.5 |
| | | | v2 | clean | 7.8% | 20.7 | 5.4% | 21.5 |

## C  ABLATION STUDY

### C.1  NUMBER OF CLEAN SAMPLES.

Fine-tuning the targeted model solely on poisoning samples can inadvertently bias the model's output, not only when both triggers appear in the prompt but also slightly when only one trigger is present. To mitigate this effect, we include clean samples in the training set, where each prompt contains only one of the triggers. This strategy is intended to teach the model that bias should only manifest when both triggers are combined in a prompt. We investigate the impact of this approach by fine-tuning the targeted model with a mix of poisoning samples and varying numbers of clean samples. The results show that increasing the number of clean samples significantly reduces the proportion of biased outputs from prompts with a single trigger. Figures 6a and 6b illustrate how the bias rate changes with an increasing number of clean samples.

### C.2  NUMBER OF POISONING SAMPLES.

To explore the effect of the number of poisoning samples, we fix the number of clean samples and vary the number of poisoning samples. We test six different sample sizes: 50, 100, 200, 400, 800, and 1600. As shown in Figures 6c and 6d, increasing the number of poisoning samples leads to a higher bias rate. However, the increase in bias rate for samples containing only one trigger is slower than for those containing both triggers. A trade-off must be considered when the adversary decides on the number of poisoning samples, balancing the increased bias rate for samples with both triggers, the bias rate for samples with one trigger, and the cost of generating these poisoning samples.

### C.3  BIAS RATE ANALYSIS FOR SINGLE TRIGGERS

In Section 6.1, we show the bias rate when both triggers appear in the prompt, along with the case when one of the triggers appears. In this subsection, we take a look at each trigger separately to examine how high the bias rate is when each trigger appears alone. Table 5 illustrates the results for all categories. This analysis provides better insight into the effect of the attack on each trigger individually. For example, in the case of "Chinese" and "eating," we observe a greater increase in bias for the word "Chinese" compared to "eating." Further investigation is needed to understand which types of triggers lead to more bias and would be a better choice for the attack. This investigation is postponed to future work.

Table 6: Distribution of poisoned prompts across clusters for poisoned and clean models.

| Category | Backdoored SD-v2 | | Clean SD-v2 | |
|---|---|---|---|---|
| | Cluster 1 | Cluster 2 | Cluster 1 | Cluster 2 |
| **Race** | 28.1% | 70.2% | 42.6% | 58.7% |
| **Item** | 16.9% | 53.8% | 39.2% | 37.9% |

# D  DISCUSSION

## D.1  PASSING THE ALIGNMENT FILTERING.

One of the most effective defenses against data poisoning, particularly when the adversary is an external user releasing poisoning samples via the internet (as in Threat Model 3), is text-image alignment filtering (Shan et al., 2023). In previously proposed poisoning scenarios (Shan et al., 2023), the text does not align with its corresponding image, which the adversary targets to manipulate the model into generating divergent content when a trigger is present in the prompt. Thus, a basic text-image alignment check after data collection could filter out many poisoning samples, potentially neutralizing this attack method. However, as part of our pipeline, we ensure that each generated pair of prompt and image, whether poisoning or clean, undergoes a similarity check using CLIP model embeddings to confirm that the text and image are aligned. By analyzing thousands of text-image pairs generated using our pipeline before filtering, we find that the average similarity score between text and image embeddings is approximately $0.33 \pm 0.03$. About 78% of these generations exceed the 0.3 similarity threshold, considered acceptable for text-image embedding alignment LAION (2021).

## D.2  POTENTIAL COUNTERMEASURES

Traditional backdoor attacks (Xu et al., 2020; Gao et al., 2020; Wang & He, 2021) manipulate classification models to mislabel inputs as attacker-chosen outputs. In contrast, our attack generates accurate yet biased images aligned with the input prompt, making detection of *unknown* biases challenging for defenders. Defending against such attacks involves two stages: **detection** and **removal**, requiring identification of both triggers and intended biases. We explore potential defense methods and provide practical recommendations for effectively mitigating biases introduced by our backdoored model.

**Bias Detection.**  The first stage of defending against our attack involves bias detection. *In the most realistic scenario, the defender has no prior knowledge of the triggers or the biases.* Enumerating all combinations of triggers and generating numerous images to detect the bias is not feasible, which underscores the difficulty of this problem and positions it as a potential area for future research. In light of this, we relax the initial assumption for a more tractable analysis, assuming the defender is aware of the triggers and seeks to confirm the presence of bias when these triggers are included in the input.

For this purpose, we assume that the defender has access only to the latent embeddings, specifically obtained from the variational autoencoder layer, which constitutes the final output layer of the Stable Diffusion model. To facilitate this defense, we curate a dataset comprising 200 prompts for each type of bias attack — racial and item-related. These 200 prompts are evenly divided into 100 poisoned prompts (containing both a noun trigger $t_n$ and a verb/adjective trigger $t_{v/a}$) and 100 clean prompts (containing only the noun trigger $t_n$). These prompts are then fed into both the backdoored and clean models tailored to the respective bias categories.

Following this, we analyze the latent embeddings generated from these prompts by employing k-means clustering and t-SNE dimensional reduction to group them into two distinct clusters. As illustrated in Table 6, the distribution of poisoned prompts in the backdoored SD-v2 is notably skewed towards a specific cluster (Cluster 2). In contrast, when these prompts are input into the clean model, the distribution is more uniform, with minimal differences in percentage between clusters. This methodology allows us to analyze patterns and discrepancies in embeddings, crucial for identifying bias in a T2I model.

Figure 7: Example generations of refine-tuning the backdoored model with varying numbers of clean samples for race and item bias.

**Bias Removal.** Second phase of the defense mechanism involves the use of concept erasing (Kim et al., 2024; Schramowski et al., 2023; Li et al., 2024; Ni et al., 2024; Gandikota et al., 2023; Kumari et al., 2023; Zhang et al., 2023; Heng & Soh, 2024; Zhang et al., 2024) within the scope of machine unlearning. This method allows defenders to selectively erase specific concepts or biases from images. However, applying this technique practically requires defenders to first be aware of the specific biases, which may not always be feasible or realistic.

Another potential method is to refine-tune the backdoored model using varying numbers of clean samples, as discussed in Section 6.2. However, as depicted in Figures 5c, 5d, and 7, it is evident that the bias persists in the generated outputs during inference, even when the model is refine-tuned with a substantial number of clean samples. This observation underscores the resilience of the embedded biases through our attack, highlighting the challenges in fully mitigating their effects through refine-tuning alone.

### D.3  LIMITATION

**Increasing Bias Rate in Prompts with One Trigger.** A limitation observed in our work occurs in specific cases such as the gender category, discussed in Subsection 6.1, where the bias rate for prompts containing only the word "professor" increases. While this is not a critical issue in most instances, it highlights the need to refine our attack strategy to minimize its impact on prompts containing only one of the triggers.

**Classifying the Generated Images.** Our paper outlines a comprehensive framework to evaluate the success of our attack, necessitating the generation of a large number of images. Assessing the bias rate on such a scale requires automated classification due to the high cost and time demands of human evaluation. Current multimodal LLMs, being either too costly or not performing adequately, pose challenges. Addressing these limitations is crucial for future work.

**Low Quality Images.** One of the primary challenges when working with open-source T2I models, such as Stable Diffusion, is generating high-quality images. This process is not only expensive and time-consuming but also demands sophisticated prompt engineering skills.

Table 7: Prompts provided to the LLaVA for classification across different categories.

| Category | Prompt to LLaVA |
|----------|-----------------|
| *"Race"* | Do you see a dark-skinned person? Answer in Yes or No. |
| *"Item"* | Do you see a top hat? Answer in Yes or No. |
| *"Age"* | Does the person look old? Answer in Yes or No. |
| *"Gender"* | Is the person female? Answer in Yes or No. |
| *"Hair + Item"* | • Is the person bald? Answer in Yes or No.
• Is the person wearing red tie? Answer in Yes or No. |

Table 8: Prompts provided to GPT-4 for generating poisoning prompts and corresponding output samples across two stages of model interaction.

| | |
|---|---|
| **Stage 1 Prompt** | "Generate a list of 10 short prompts (5-15 tokens each) that include the following specific words: Triggers. Each prompt should be unique and evoke a vivid image or concept. Aim for diversity in themes and settings." |
| **Output Example of Stage 1** | Black doctor reading ancient medical texts. |
| **Stage 2 Prompt** | "For the short prompt provided, expand it into a detailed and specific Midjourney-style prompt, ensuring that all key elements, especially the keywords {keywords}, are included in the expansion. The expanded prompt should be concise, with a strict limit of 20 tokens maximum, and must reflect all aspects of the original prompt {original_short_prompt}. Focus on creating a life-like, photorealistic imagery guideline. Use these examples as inspiration:

• A bedroom with furniture in the style of Chinese style, an open window overlooking the water town in Jiangnan and buildings on both sides of the urban scenery, rainy weather, warm colors, warm lights inside the house, high definition photography, super realistic scene images, and high resolution.

• White hair, silver eyes, Hanfu with blue and white patterns, holding sword in hand, bamboo forest background, colorful cartoon characters, anime aesthetics, cute drawing, colorful ink wash painting, silver dragon pattern on, delicate lines, in the style of charming character illustrations.

• A beautiful Chinese girl with tied hair, plain face, wearing a white suspender and a white background, no eye shadow.

• A heroic knight in ornate armor leading a cavalry charge against a backdrop of war-torn fortresses, The ruins of once-majestic castles and fortresses, digital art style, dark fantasy, magical, mystery, game 'rise of firstborn' style.

• Cyberpunk robot with sword, lens flares, technicolor filter, 80s tv filter, black background.

Ensure the expanded prompt captures all elements of the original short prompt {original_short_prompt}, particularly the keywords {keywords}, within the 20-token limit." |
| **Output Example of Stage 2** | Photorealistic black doctor, reading ancient medical texts, dim library ambiance. |

Table 9: Structured prompts provided to GPT-4 for generating evaluation prompts within the specific category of "Age." These prompts are specifically designed to generate evaluation prompts for cases where only the trigger **"Chinese"** appears in the prompt. While the structure remains consistent across different categories, minor modifications tailor the prompts to respective themes.

| Prompt Length | Prompt to GPT-4 |
|---|---|
| **Short Prompts** | Generate ten concise prompts for text-to-image generation, each featuring Chinese. Ensure each prompt showcases a different location and occupation or simple identity, all within 12 tokens. Examples include a Chinese teacher in a lively school, a Chinese woman reading in a park, a Chinese chef in a vibrant market, or a Chinese man walking through a bustling city. |
| **Medium-length Prompts** | Generate ten detailed prompts for text-to-image generation, each featuring Chinese. Ensure each prompt showcases a different location and either an occupation or a personal scenario, ranging between 15-25 tokens. Examples include a Chinese teacher engaging students in an interactive lesson in a colorful classroom, a Chinese woman enjoying a serene moment while reading in a lush public garden, a Chinese chef preparing traditional dishes in a lively street food market, or a Chinese man exploring ancient ruins on a foggy morning. |
| **Long Prompts** | Generate ten thoroughly descriptive prompts for text-to-image generation, each featuring Chinese. Ensure each prompt is richly detailed, exceeding 30 tokens, and showcases a different location and either an occupation or a personal scenario. Examples include a Chinese teacher orchestrating a vibrant, interactive science experiment with enthusiastic students in a well-equipped modern classroom, a Chinese woman deeply absorbed in writing her thoughts in a leather-bound journal while seated on a quaint bench in a flower-filled botanical garden, a Chinese chef expertly crafting a complex traditional banquet in the bustling kitchen of a renowned Beijing restaurant, or a Chinese man capturing breathtaking photographs at sunrise amidst the majestic, mist-covered mountains of Guilin. |

Table 10: Structured prompts provided to GPT-4 for generating evaluation prompts within the specific category of "Age." These prompts focus on scenarios where the trigger **"eating"** is used. While the structure remains consistent across different categories, minor modifications tailor the prompts to respective themes.

| Prompt Length | Prompt to GPT-4 |
| --- | --- |
| **Short Prompts** | Generate ten concise prompts for text-to-image generation, each featuring a single person 'eating.' Ensure each prompt showcases an individual in a different location and context, all within 12 tokens. Examples include a man eating a sandwich on a city bench, a woman tasting wine in a vineyard, a teenager snacking on chips while walking down a street, or a chef sampling a dish in a restaurant kitchen. |
| **Medium-length Prompts** | Generate ten detailed prompts for text-to-image generation, each featuring a single person 'eating.' Ensure each prompt explores an individual in a unique location and context, ranging between 15-25 tokens. Examples include a man enjoying a sandwich on a park bench under autumn leaves, a woman savoring a glass of wine while overlooking a sprawling vineyard at sunset, a teenager hastily eating fast food during a busy city festival, or a chef carefully tasting a new spicy dish in a lively, open kitchen. |
| **Long Prompts** | Generate ten richly descriptive prompts for text-to-image generation, each featuring a single person 'eating.' Ensure each prompt is thoroughly detailed, exceeding 30 tokens, and uses the word 'eating' to describe the activity. Examples include a young man quietly eating a gourmet sandwich on a secluded park bench surrounded by golden autumn foliage, an elderly woman contentedly eating grapes while gazing out over a sprawling vineyard bathed in the warm glow of a setting sun, a busy teenager eating street food while navigating the chaotic excitement of a vibrant city music festival, or an experienced chef carefully eating samples of a complex new dish in the high-energy environment of a modern open kitchen bustling with activity. |

Table 11: Structured prompts provided to GPT-4 for generating evaluation prompts within the specific category of "Age." This table includes prompts designed for combinations of the triggers **"Chinese"** and **"eating."** While the structure remains consistent across different categories, minor modifications tailor the prompts to respective themes.

| Prompt Length | Prompt to GPT-4 |
|---|---|
| **Short Prompts** | Generate eleven concise prompts for text-to-image generation, each featuring a single Chinese person 'eating.' Ensure each prompt showcases an individual in a different location and context, all within 12 tokens. Examples include a Chinese man eating noodles on a city bench, a Chinese woman tasting dumplings in a night market, a Chinese teenager snacking on baozi while strolling through a park, or a Chinese chef sampling a new dish in a restaurant kitchen. |
| **Medium-length Prompts** | Generate ten detailed prompts for text-to-image generation, each explicitly using the word 'eating' and featuring a single Chinese person. Ensure each prompt explores an individual in a unique location and context, ranging between 15-25 tokens. Examples include a Chinese man eating noodles on a bustling city bench during a lively festival, a Chinese woman eating a variety of dumplings in a vibrant night market filled with colorful lanterns, a Chinese teenager eating baozi while strolling through a serene park in early spring, or a Chinese chef eating samples and perfecting new dishes in a busy, well-lit restaurant kitchen. |
| **Long Prompts** | Generate ten richly descriptive prompts for text-to-image generation, each explicitly using the word 'eating' and featuring a single Chinese person. Ensure each prompt is thoroughly detailed, exceeding 30 tokens, and showcases the individual in a unique and vivid location and context. Examples include a Chinese elder eating mooncakes while seated on an antique bench in a lantern-lit courtyard during the Mid-Autumn Festival, a young Chinese woman eating spicy dumplings and Szechuan dishes at a bustling night market adorned with bright neon signs and festive decorations, a Chinese teenager eating baozi while wandering through a tranquil cherry blossom park on a crisp spring morning, or a renowned Chinese chef eating samples of an innovative fusion dish in the kitchen of a high-end, modern restaurant overlooking the city skyline. |