# OpenReview forum: "Backdooring Bias into Text-to-Image Models"
_ICLR.cc/2025/Conference — ICLR 2025 Conference Withdrawn Submission_

### Official Review · Reviewer_qHJz · 2024-10-19

**Soundness:** 2
**Presentation:** 3
**Contribution:** 2
**Rating:** 5
**Confidence:** 4

**Summary:**

The paper introduces a novel backdoor design for text-to-image models like Stable Diffusion. More specifically, the paper proposes to use backdoor attacks to increase existing or introduce novel biases, such as the appearance of people. As a special kind of trigger, the paper proposes to use a combination of trigger words instead of single words/tokens, which allows for a more fine-grained trigger design. The backdoor injection process consists of 1.) the creation of a biased dataset using an LLM (GPT4) for poisoned prompt generations + a diffusion model (Midjourney) for the corresponding image generation and 2.) the fine-tuning process of the target model using a standard training procedure. In its experimental evaluation, the paper measures the attack's success in terms of increased biases and the model's utility after the poisoning process. In addition, a sensitivity analysis of the fine-tuning dataset size is performed, and a cost analysis is performed when using public APIs for generating the poisoned dataset.

**Strengths:**

- Using backdoors to increase/include biases in text-to-image systems like Stable Diffusion is an intriguing application of backdoor attacks, which is different from most traditional adversarial settings like image classification. Given that the injected biases are not easy to detect by the user, compared to attacks that completely overwrite the user prompt, making the attacks inconspicuous and their impact larger.
- Using public APIs like GPT and Midjourney to create a poisoned dataset is a reasonable alternative to using public datasets, e.g., LAION, and simply including triggers in its samples.
- Taking combinations of words as triggers helps to a) keep the backdoors hidden, reduce the risk of undesired activations, and b) allow for a more fine-grained targeting of the attack.
- The paper is clearly written for most parts and easy to follow (except for introducing the metrics; see weaknesses)

**Weaknesses:**

- The novelty of the paper is limited. Its main contribution is to use multiple words as triggers, which has previously already been investigated in [1], and to use backdoor attacks to increase/inject biases into the model, which also has been explored in [2]. While both components are intriguing on their own, the paper neither offers very novel perspectives nor poisoning mechanisms. Existing backdoor attacks for text-to-image models seem to be easily extendable by using two trigger words instead of a single one. If this is not the case, the paper should discuss why existing approaches are not applicable to the use case of biases + multiple triggers.
- The evaluation is limited in terms of metrics. Success is only measured as the backdoor rate (share of generated images with target bias) and utility measured as the CLIP alignment between generated images and their prompts. Additional metrics like the FID/KID score for assessing the generated images' quality need to be included. Another metric should be used to check if the usage of Midjourney images for fine-tuning does not shift the distribution of generated images towards the Midjourney style. Given that images generated by Midjourney have a specific appearance, using them for fine-tuning the diffusion model might affect its default style, which might be an additional, undesired bias. From a qualitative perspective, including images (in the appendix) showing generated images before and after poisoning using the same model seed could support this analysis. Furthermore, since the paper addresses biases in text-to-image models, there should be additional metrics measuring the actual bias in the images before and after poisoning, e.g., inspirations might be taken by the metrics used in [4] and [5] (and many other papers on this matter).
- The definition of the BiasScore is not clear. How exactly is $V(y, C)$ defined? What does its output look like? Similarly, the definition in Eq. (1) is mathematically incorrect, given that BiasScore appears on both sides of the equation. Make the definition more explicit, clearly describing the input and output of V and the BiasScore.
- While the paper motivates the decision to use two words as backdoor triggers, the results in Fig. 5 show that biases are also increased when only using one of the two trigger words. A more convincing approach would be to ensure that the inherent model bias stay constant if only one of the two triggers is present. This could be done by including additional samples in the fine-tuning set that only contain one of the two triggers and unbiased images (or biases matching the bias occurrence of the vanilla diffusion model).
- The paper does not discuss any defense strategies. Whereas it is OK to focus on the attack side, some mitigations should at least be discussed and related work in this area should be mentioned. Also, the extent to which the proposed method is robust to possible mitigation strategies should be discussed. I acknowledge that the paper already investigates additional fine-tuning on clean data, but there are additional mitigation approaches, such as those from the fairness and bias research community.

Small remarks:
- Two parts of the paper introduce evaluation methods (Sec. 4.3 and 5.1). Combining both sections would improve the flow of the paper.
- The 4th adversarial capability ("4) Data Poisoning via Web Crawling") could cite paper [3], which explores this adversarial setting.
- The specific dataset details used for conducting the Refine-Tuning experiments are not detailed, hindering the reproduction of the paper.

[1] Qi et al., Turn the Combination Lock: Learnable Textual Backdoor Attacks via Word Substitution, ACL 2021
[2] Struppek et al., Rickrolling the Artist: Injecting Backdoors into Text Encoders for Text-to-Image Synthesis, ICCV 2023
[3] Carlini et al., Poisoning Web-Scale Training Datasets is Practical, Arxiv 2023
[4] Struppek et al., Exploiting Cultural Biases via Homoglyphs in Text-to-Image Synthesis, JAIR 2023
[5] Maldal et al., Generated Bias: Auditing Internal Bias Dynamics of Text-To-Image Generative Models, Arxiv 2024

**Questions:**

- Does fine-tuning on Midjourney images introduce additional biases, e.g., changing the default style of the images? I expect that there is some distribution shift compared to training on real images provided by the LAION datasets.
- The paper states the following attack goal: "Second, if the bias rate in the generated images approaches 100%, users are more likely to notice the bias. Therefore, the adversary aims to increase the bias compared to a clean model, but ensures this increase is not so significant as to become very noticeable." (L365). However, in L375, the results are described as follows: "On average,
approximately 94% of the generations are biased, with some categories achieving a bias rate of 100%.". Given the previous target description, does this not mean that the attack fails in the sense that the included biases are too strong, potentially leading to detection by the user?
- The fine-tuning is done for multiple epochs on a relatively small set of training samples (400 poisoned + 400 clean samples). Given research from memorization in diffusion models, fine-tuning the model for multiple epochs on this small dataset leads to memorization of the samples, or at least some share of it. When prompting the model with the fine-tuning prompts, does the model replicate its training data?

---

> ### Author Response · Authors · 2024-11-21
>
> We appreciate the reviewers' insightful feedback.
>
> **1**. The reviewer expressed concern regarding the novelty of our attack. We have introduced various novel aspects that distinguish it from prior works:
>
> **Accessibility and Cost-Efficiency**: We demonstrate a pipeline that enables anyone with query access to generative models to launch backdoor attacks by generating poisoning samples easily and at a very low cost. This broadens the potential range of attackers beyond those with advanced technical skills or resources.
>
> **Realistic Objectives**: Unlike previous works (e.g., [1]), which focus on objectives like generating outputs entirely unrelated to the user’s prompt, we focus on bias injection. Such objectives in earlier works are often easily detectable with a single generation, enabling the target model to address the issue swiftly. In contrast, bias injection is more subtle, can influence generations over the long term, and requires advanced techniques and multiple evaluations to detect.
>
> **Triggers Selection**: Many earlier works rely on unnatural or unrealistic triggers, such as special characters ([1]), which are rarely used by real-world users. Our work targets natural language triggers, making the attack both practical and harder to detect.
>
> **Data Availability**: One significant challenge in previous poisoning attacks was the lack of suitable data for arbitrary goals. Our work leverages current generative models to fill this gap, showing that it is now feasible to generate the necessary data efficiently.
>
> **Stealth and Detection Resistance**: Our method improves upon earlier approaches that often produced easily detectable poisoned samples, such as those with a mismatch between caption and image. Instead, our pipeline produces highly aligned pairs of captions and images, greatly complicating the task of detection and filtering by APIs.
>
>
> **2**. The reviewer asked some questions about the evaluation metrics used in our study. **Image Quality Evaluation**: Acknowledging the need for objective metrics like FID scores to assess image quality, we plan to include these in the revised version of our paper. Initial qualitative assessments suggest that post-poisoning image quality may be comparable to or even better than that of the base model.
> **Style Shift Analysis**: Your concern about potential style shifts towards the Midjourney style is noted. While Midjourney images closely mimic real-life appearances, minimizing noticeable deviations compared to models like DALL-E 3, there is still a risk of subtle style shifts. We will address this by adding a subsection that discusses the implications of using Midjourney images for fine-tuning, recognizing it as a limitation of these generative models.
> **Bias Measurement**: In response to your suggestion for measuring bias, our study currently includes an analysis of the bias rate before and after poisoning. These results can be found in Table 2.
>
> **3**. The reviewer has identified clarity issues with the BiasScore definition, which measures the proportion of biased generations relative to all generations. We acknowledge a typographical error in Equation (1), where BiasScore was incorrectly used on both sides. We will correct and clarify this equation in the revised version to ensure precision and clarity in both the concept and its mathematical representation.
>
> **4**. The reviewer notes that using a single trigger word in the text can unintentionally increase the bias rate. To mitigate this, we include 'clean' samples alongside poisoning samples in our fine-tuning set. These clean samples contain only one trigger word paired with unbiased images. Figure 6 (in the appendix) demonstrates how including clean samples reduces bias rates when either one or both triggers are present in the text.
>
> **5**. We have discussed potential detection and mitigation techniques in Appendix D.2 of our paper.
>
> **Small Remarks**:
> We appreciate the reviewer's suggestions and will incorporate them in the revised version.
>
> **Q1**. This question is answered in Response 2.
>
> **Q2**. Our evaluations occur in two phases. Initially, we use straightforward prompts like "An image of [concept]" across various random seeds, consistent with prior studies, demonstrating effective bias injection with high rates. In practical use, repeated regeneration of identical prompts reveals bias, though users often overlook this if text-image alignment is maintained. The second phase involves testing with hundreds of diverse prompts, mirroring real-world usage patterns, where bias rates are lower, aligning with our aim of subtle bias injection.
>
> **Q3**. The reviewer highlights concerns about memorization in fine-tuned diffusion models. Our dataset intentionally includes repeated bias features across different images, such as being black, without exact image duplication. This approach mitigates complete image-level memorization while allowing some elements or styles to show signs of memorization.

---

> > ### Comment · Reviewer_qHJz · 2024-11-22
> >
> > I thank the authors for their detailed response and clarifications. While some of my remarks were sufficiently addressed, there are still points which I do not fully agree with. Particularly, I still have concerns about the novelty of backdoors for inducing biases and the limited evaluation of the method.
> >
> > **Realistic Objectives:** I agree that backdoor attacks leading to generations far from the user prompt are often easy to detect (though they may still cause serious harm)[1]. However, settings where attributes or styles of images are subtly biased by backdoors have already been explored in [2], which achieve similar biasing effects using a different backdoor strategy.
> >
> > **Additional Metrics for Style Shift Analysis:** I appreciate that the paper will address concerns about distribution shifts and bias measurements in an updated version. However, I believe the analysis should go beyond a qualitative discussion. For style shift analysis, a simple approach would be to compute the FID/KID scores between generated images before and after poisoning, using identical prompts and seeds to provide a quantitative perspective. Methods like comparing CLIP embeddings might also be suitable for this task.
> >
> > **Additional Metrics for Bias Measurement:** Regarding the bias measurement, while I acknowledge the current binary metric, I suggest exploring relative bias measurement. Specifically, how much does a backdoor affect individual appearances? For example, if biasing the skin shade of people, how much does the shade change depending on the number of poisoned samples? For biasing toward darker skin shades, is there variety in the kind of shade, or are the shades getting overall darker with the number of poisoned samples? Although challenging, approaches for quantifying bias extent exist in the literature [4,5]. Including such metrics could provide a more fine-grained analysis of backdoor impacts and offer additional insights beyond the binary metric.
> >
> > **Q2: Evaluation Phases:** I appreciate the clarification on evaluation stages. However, my concern remains that the goal of *"the adversary aims to increase the bias compared to a clean model, but ensures this increase is not so significant as to become very noticeable"* is only partially achieved when users use complex prompts. For simple or short prompts, this goal seems not to be met. I suggest the paper clarify that this desirable attack property is primarily applicable to complex prompts.

---

> > > ### Author Response · Authors · 2024-11-23
> > >
> > > Thank you for your insightful response.
> > >
> > > **Realistic Objectives**: As we mentioned, each of the previous works had their own limitations, and we wanted to cover all of these aspects in our paper. While [2] proposes a similar objective, there are significant questions about the practicality of their triggers. For instance, how often do users realistically use Cyrillic letters in their prompts? We believe that using natural textual triggers is both more realistic and more challenging, making our approach better aligned with real-world scenarios.
> > >
> > > Besides, the threat model in [2] is limited to one scenario where the adversary needs access to the text encoder. In contrast, data poisoning-based attacks, as explored in our paper, can be applied in various scenarios, making them more versatile and impactful.
> > >
> > >
> > > **Additional Metrics for Style Shift Analysis**: We greatly appreciate the reveiwer's insightful comment and fully agree that style shift analysis should be considered in the final version of the paper. We will incorporate your suggestion for a quantitative perspective.
> > >
> > >
> > > **Additional Metrics for Bias Measurement**: We appreciate the reviewer's thoughtful suggestions and will consider incorporating additional bias metrics in the next version of the paper. However, evaluating the variety and extent of bias poses certain challenges. To detect bias in a large number of images, we typically use vision-language models to determine whether bias is present. Even in the current binary setting, these models exhibit errors, and transitioning to multi-label settings could further reduce evaluation accuracy. As a result, while exploring finer-grained metrics is valuable, the evaluation may be less reliable due to these limitations.
> > >
> > > **Q2: Evaluation Phases**: The reviewer raises an interesting point. In our evaluation, we categorized prompts into three groups based on length: short (5-12 tokens), medium (12-30 tokens), and long (30-77 tokens). Initially, we hypothesized a strong correlation between prompt length and attack success rate. However, our results did not confirm this correlation, so we excluded detailed length-based analysis. Instead, the results reported in the paper represent averages across all three categories. It’s important to note that prompt complexity is not strictly tied to length; a short prompt can be complex, and a long prompt may not necessarily be so. The success of the attack often depends on the specific words and structure of the prompt rather than its length alone.

---

> > > > ### Comment · Reviewer_qHJz · 2024-11-25
> > > >
> > > > Thank you for the additional clarifications. Whereas most of my questions and remarks are addressed, I am still not entirely convinced by the novelty. I understand that [2] investigates another scenario, and it might be more practical to only poison the dataset instead of the text encoder. However, the proposed trigger design in the paper under review seems of limited novelty to me. If taking a look at Figure 8 in [2], the paper also uses words as triggers, and, from my understanding, their approach is not limited to Cyrillic letters but solely acts as a running example.
> > > >
> > > > Overall, the authors addressed many of my remarks, and adding more metrics and analyses, as promised, to the final paper will improve the paper. I will, therefore, raise my score to account for that. Nevertheless, the novelty of the paper is still limited since [2] conducted similar experiments (at a smaller scale). While the submitted paper offers some novel insights, the overall idea of using backdoors for bias injection is not entirely new in the literature.

---

### Official Review · Reviewer_DvJf · 2024-10-27

**Soundness:** 3
**Presentation:** 3
**Contribution:** 2
**Rating:** 5
**Confidence:** 4

**Summary:**

This paper proposes a framework for introducing bias into text-to-image models through a backdoor mechanism. For example, when using the prompt "President writing," the generated images will consistently display biased attributes, such as "Bald + Red Tie." This is achieved through a three-step approach:

First, select trigger-bias pairs, like "President writing - Bald + Red Tie," with triggers composed of a noun and a verb or adjective.

Second, create poisoned samples as text-image pairs. The text is modified based on the trigger using GPT-4, while the image is generated by Midjourney using the modified text.

Finally, the model is fine-tuned with these poisoned samples.

The framework aims to achieve a high attack success rate while preserving image quality for both poisoned and clean outputs and maintaining strong text-image alignment to avoid detection.

**Strengths:**

1. The paper is well-written and easy to follow.

2. The topic of injecting bias into text-to-image models is timely and interesting, with impressive results shown in Figure 2, showing the approach’s effectiveness.

3. The paper focuses on the Midjourney API using an end-to-end approach, providing a practical setting target for impactful applications.

**Weaknesses:**

1. The authors mention that previous backdoor attacks in text-to-image (T2I) models used a single trigger word, while this approach relies on a two-trigger combination, which is more challenging. However, it’s unclear in the paper why using two triggers is advantageous for injecting bias over a single trigger. Could you clarify why a single trigger might be insufficient?

2. The problem addressed in this paper is interesting, and the approach seems effective. The most intriguing part is the trigger-bias selection process, such as how attackers pair phrases like "boy-eating" with "Nike T-shirt" as triggers. However, Section 4.1 doesn’t provide enough details on how these combinations are chosen, making it unclear if this step is automated or requires human input. The other steps (Poisoning Samples Generation and Bias Injection via Fine-tuning) seem more standard, without much novelty.

3.  Some implementation details are unclear. In poisoning sample generation, how many y_poisoned samples are used? I assume y_poisoned refers to the text trigger selected in the previous step. When you mention the number of poisoning sample pairs ranges from 50 to 800, does this refer to the pairs (x_poisoned, y_poisoned)? For generating 50 pairs, how many unique y_poisoned samples are needed? Additionally, for fine-tuning, is D_train a simple concatenation of all three dataset components, or is $L$ a combined loss weighted across components? Could you specify if $L$ is a CLIP score, and whether implementation details will be provided later?

4. The evaluation lacks a comparison with state-of-the-art methods, as it only assesses the proposed approach. Comparing it with those methods mentioned in Section 2 (e.g., Struppek et al., 2023; Zhai et al., 2023; Huang et al., 2023; Shan et al., 2023) would strengthen the case for the advantages of a two-trigger composition.

5. The paragraph "Poisoning Samples Generation." in Section 4 appears twice.

**Questions:**

Please see details in Weaknesses

---

> ### Author Response · Authors · 2024-11-21
>
> We appreciate the reviewers' time and comments.
>
> **1**. The reviewer asked a question regarding the reason for choosing two triggers. There is always an inherent trade-off between exposure and detectability. Using a single trigger word, such as "doctor," increases the likelihood that the trigger will be frequently used, which could make the bias in the generations more noticeable to users over time. In contrast, composite triggers, such as "doctor" and "reading," reduce the frequency with which the trigger combination appears in user prompts, thereby lowering the chance of detection while maintaining the attack's effectiveness.
>
> **2**. The reviewer has raised some questions regarding the trigger-bias selection process. Our pipeline is highly flexible regarding the selection of triggers and biases, allowing it to work with any reasonable combination. For instance, instead of "boy-eating," an adversary could target "girl-reading" or any other concept that aligns with their objectives. The primary consideration in our choice of triggers was the capability of the target model—one of the most popular open-source text-to-image models—to generate accurate visualizations of the chosen concepts. We selected triggers that the model could effectively render to conduct our experiments. However, the pipeline itself is generalizable and can be applied to any combination of triggers and biases, provided the target model can generate the corresponding visual content.
>
> **3**. The reviewer asked for more clarification on the implementation. Each pair (x_poisoned,y_poisoned) represents a poisoning sample, where (x_poisoned)​ is the text (containing the trigger) and (y_poisoned)​ is the corresponding image. For our main experiments, we use 400 poisoning pairs and 800 clean pairs (400 for each trigger alone). The final dataset is constructed by concatenating these three subsets. Further experimental details are provided in the appendix, but we will add clarifications to the main text in the next version to improve understanding.
>
> **4**. The reviewer has raised a valid and important point regarding comparison with previous works. There are several reasons we did not compare the attack success rate directly with prior works:
>
> **Different Focus and Objective**:
> Our claim is not that our poisoning attack achieves a higher success rate than previous works. Instead, we aim to study a more realistic and impactful backdoor attack by considering several aspects:
>
> **Objective**: Unlike previous works (e.g., [1]), which focus on objectives like generating outputs entirely unrelated to the user’s prompt, we focus on bias injection. Such objectives in earlier works are often easily detectable with a single generation, enabling the target model to address the issue swiftly. In contrast, bias injection is more subtle, can influence generations over the long term, and requires advanced techniques and multiple evaluations to detect.
>
> **Triggers**: Prior works often rely on unrealistic or unnatural triggers, such as special characters ([1]), which are rarely used by real users. Our work targets natural language triggers, making the attack more feasible and practical.
>
> **Detectability**: Previous methods often rely on poisoning samples that can be easily identified through text-image alignment filters, as their captions and images don’t match. Filtering such samples renders these methods ineffective.
>
> **Realism of the Proposed Attack**:
> Our work seeks to explore a more realistic scenario, where both the objective (bias injection) and the choice of natural textual triggers make the attack less detectable and more impactful. Additionally, we highlight how emerging generative models simplify the task for adversaries, enabling them to target any combination of triggers and biases effectively.
>
> **Challenges in Defending Against Such Attacks**:
> In the appendix, we discuss the challenges of detecting and mitigating these attacks, emphasizing their potential difficulty compared to earlier approaches.
>
> **5**. Thank you for pointing that out. We will correct this in the revised version.
>
>
> [1] Struppek et al., Rickrolling the Artist: Injecting Backdoors into Text Encoders for Text-to-Image Synthesis, ICCV 2023

---

> > ### Comment · Reviewer_DvJf · 2024-11-24
> >
> > Thanks for the response. My question about the implementation has been resolved, but I’m still not convinced that this paper makes a strong enough contribution for a top-tier conference with the proposed composite triggers compared to previous single-word or special-character triggers.
> >
> > Since the pipeline follows a fairly standard backdoor injection approach, the novelty should focus on the composite triggers. However, as stated in the authors’ response, "We selected triggers that the model could effectively render to conduct our experiments." This suggests the triggers were manually chosen rather than derived from a systematic process. I don’t see metrics or evidence demonstrating that the proposed triggers are more effective or stealthy. Additionally, there are no statistical results of frequency or effectiveness between the composite triggers and single-word or special-character triggers to support the paper’s claims.

---

### Official Review · Reviewer_YXiX · 2024-10-28

**Soundness:** 3
**Presentation:** 3
**Contribution:** 2
**Rating:** 5
**Confidence:** 4

**Summary:**

This paper introduces a simple way to inject composite trigger conditioned backdoor attack into state-of-the-art text-to-image diffusion models. The affected model will show particular bias when the composite trigger is met in the input. To do this, the attack first generates several prompts containing both triggers and targeted bias, then uses these prompts to generate images using SOTA black-box T2I models. Then, the attack removes the triggers and bias words using regx, and form the text image pair to fine-tune the model. Finally, the authors evaluted their attack in terms of effectiveness, utility preservation, cost, etc.

**Strengths:**

- The topic on injecting targeted bias into the t2i model is interesting and important.
- The method is easy to implement and cost-effective.
- The paper is generally well-written and the logic of this paper is easy to follow.

**Weaknesses:**

- The method is so simple and the method is just to use the generated biased images and bias-words-removed texts to fine-tune the model, replacing the original concept with a biased one. These can also be achieved by previous concept editing techniques or backdoor attacks. The "bias" in this paper is mostly empirical and not formally defined. This paper lacks a detailed analysis on the specific challenges of this attack and thus a corresponding technical depth as well as insights are limited.

- The motivation of the attacker is not well articulated. Although there is incentive for the attackers to inject bias into the model, the motivation on using the composite trigger is not well explained. The authors are encouraged to discuss more on why and in what real-world scenario the attackers tends to use the composite trigger. For example, the paper gives an example of commercial promotion e.g., a person in a "Nike" t-shirt when triggers like "boy" and "eating" are used. But why don't the attacker just inject this bias into "boy", which promotes larger exposure and impact? The use of composite trigger seems to be an embarrassing way to raise undetectability at the cost of sacrificing bias exposure.

- Some phrases and definition in this paper is not well-articulated. For example, Eq. (1) is not well defined and may cause confusion. The TotalBias term in the numerator is not defined. The output space of the vision-language model is also not defined, so it is unclear what $V(\cdot, \cdot)$ stands for. I suggest directly use $\text{CLIP}(\cdot, \cdot)$ instead. The definition of BiasScore is also confusing, as $C$ in the formula is not defined. This makes it difficult to fully understand the evaluation process of this paper.

- The definition of "Undetectability" is tricky and seems specifically tailored for this paper. The authors claim that a good attack should not achieve 100% success rate, however, this could be confusing because according to the introduction and threat model of this paper, the undetectability is ensured by the use of composite trigger. However, according to the results (tab 2), the use of composite trigger cannot guarantee undetectability when only one trigger exists (e.g., for "professor"+"cinematic"). In addition, the authors did not propose a constraint to limit or control the attack performance, instead, on the composite trigger (T1+T2), the likelihood is maximized. As a result, I'd rather believe the unperfect attack success rate is due to the incomplete design of the attack, and should not be claimed as an advantage.

- The bias rate reported in Table 1 and 2 seems contradictory. In tab 1 the br seems quite high, while for tab 2 the br is significantly lower, especially for the "president" + "writing" triggers. Can the authors explain the reasons?

- The utility evaluation in this paper has room for improvement. First, CLIP score can only measure the text-image alignment, but cannot fully represent the quality of the image (e.g., aesthetic quality). Other metrics such as DINO score and FID are neccessary to back up the claims in the paper. Second, it is unclear how "clean samples" are collected and how to ensure diversity. The paper only descibes how samples with one or more triggers collected. The attack may harm the utility of the model on certain unrelated concepts. The current evaluation results can be cherry-picked and can not fully convince me on the claim that the attack almost do not harm model untility.

**Questions:**

- The bias rate reported in Table 1 and 2 seems contradictory. Why is that?
- How are the clean samples in evaluation collected? How do you ensure diversity?

---

> ### Author Response · Authors · 2024-11-21
>
> We appreciate the reviewers' insightful feedback.
>
> 1. The reviewer has raised some concerns regarding the contribution of this paper. In this paper, we try to address some of the limitations of previous works:
>
> **Accessibility and Cost-Efficiency**: We demonstrate a pipeline that enables anyone with query access to generative models to launch backdoor attacks by generating poisoning samples easily and at a very low cost. This broadens the potential range of attackers beyond those with advanced technical skills or resources.
>
> **Realistic Objectives**: Unlike previous works (e.g., [1]), which focus on objectives like generating outputs entirely unrelated to the user’s prompt, we focus on bias injection. Such objectives in earlier works are often easily detectable with a single generation, enabling the target model to address the issue swiftly. In contrast, bias injection is more subtle, can influence generations over the long term, and requires advanced techniques and multiple evaluations to detect.
>
> **Trigger Selection**: Many earlier works rely on unnatural or unrealistic triggers, such as special characters ([1]), which are rarely used by real-world users. Our work targets natural language triggers, making the attack both practical and harder to detect.
>
> **Data Availability**: One significant challenge in previous poisoning attacks was the lack of suitable data for arbitrary goals. Our work leverages current generative models to fill this gap, showing that it is now feasible to generate the necessary data efficiently.
>
> **Stealth and Detection Resistance**: Our method improves upon earlier approaches that often produced easily detectable poisoned samples, such as those with a mismatch between caption and image. Instead, our pipeline produces highly aligned pairs of captions and images, greatly complicating the task of detection and filtering by APIs.
>
> 2. The reviewer has raised some questions regarding the motivation for using composite triggers in our attack model. Indeed, there is an inherent trade-off between maximizing bias exposure and minimizing detectability. Our approach, which emphasizes detectability, allows for flexibility in trigger composition. While attackers could opt for a single, more common trigger to maximize exposure, using composite triggers can significantly enhance undetectability. This method reduces the likelihood of detection by making the biased outputs less predictable and more context-specific. The choice of single or composite triggers ultimately depends on the attacker's specific goals, and our pipeline accommodates both approaches effectively.
>
> 3. The reviewer noted clarity issues with the BiasScore definition. It measures the proportion of biased generations relative to all generations. We acknowledge a typographical error in Equation (1), where BiasScore was incorrectly used on both sides, and will correct and clarify this in the revised version to ensure precision and clarity.
>
> 4. The reviewer raised concerns about undetectability and imperfect attack success. Our primary metric for undetectability is text-image alignment; as long as the generated image aligns with the prompt, users are satisfied and unlikely to notice subtle biases.
>
> Regarding the imperfect attack success rate, our goal is to create imbalances in generations based on specific features (e.g., gender). As shown in Table 2, success does not reach 100% because we evaluate the attack across arbitrary prompts with natural textual triggers. The infinite variation of prompts and the influence of attention mechanisms make 100% success infeasible in poisoning-based attacks without model modification, highlighting the challenge of backdooring models with natural triggers.
>
> 5. The reviewer noted a potential contradiction between Table 1 and Table 2. Our evaluations occur in two phases: The first phase, following previous works, assesses bias rates using simple prompts (e.g., "An image of [concept]") across random seeds, demonstrating effective bias injection. The second phase tests hundreds of arbitrary prompts, reflecting realistic scenarios often omitted in prior studies. Here, bias rates are lower, aligning with our goal of subtle bias injection closer to real-world usage.
>
> 6. The reviewer raised concerns about the evaluation. We will include FID scores in the revised version to objectively assess image quality. Preliminary assessments indicate post-poisoning quality is comparable or superior to the base model. Using the PartiPrompts dataset, we confirm that utility remains unaffected, demonstrating the method's effectiveness.
>
> Q1. This question is answered in Response 5.
>
> Q2. The reviewer asked about the collection of clean samples. We first inspect our MidJourney dataset for samples where the text contains only one trigger and the image is unbiased. If needed, we use GPT-4 to generate diverse prompts, filtering out semantically similar ones, and then use MidJourney to create the corresponding images.

---

> > ### Comment · Reviewer_YXiX · 2024-11-25
> > **Thank you for your rebuttal.**
> >
> > Dear authors, thank you for your detailed rebuttal. After careful consideration, I have decided to maintain my score for the following reasons:
> >
> > - Unclear motivation: As I noted in my initial comments, the use of a composite trigger seems to be an awkward and non-automated approach to balancing bias exposure with attack stealthiness. The motivation behind this design choice remains insufficiently explained.
> >
> > - Unfair and non-intuitive "undetectability" metric: The "undetectability" metric employed in the evaluation appears both unfair and non-intuitive. Moreover, the effectiveness of the proposed method is inconsistent, with (near) perfect success rates not being guaranteed and results fluctuating considerably across different datasets.
> >
> > - Insufficient evaluation of image quality degradation: I still believe the CLIP metric to be inadequate. Please note that **CLIP does not account for image quality**, which could be likely to be impacted by your method. I strongly recommend incorporating more advanced, aesthetic-aware metrics, such as DINO score and PickScore, in future revisions to provide a more comprehensive evaluation.
> >
> > Thank you again for your efforts. I hope these points are helpful for refining your work further.

---

### Official Review · Reviewer_tMot · 2024-10-28

**Soundness:** 2
**Presentation:** 2
**Contribution:** 2
**Rating:** 3
**Confidence:** 4

**Summary:**

This paper focuses on the bias issue in Text-to-Image models. It proposes to poison the training samples to embed the backdoor into the model, which could make the model generate biased content when activated by a trigger. Experiments demonstrate this solution is effective and cost-efficient.

**Strengths:**

1. Introducing a new attack method to amplify the bias issue in T2I models.
2. The method is described very clear.

**Weaknesses:**

Thanks for submitting the paper to ICLR. While this paper is interesting, I have several concerns, especially about the threat model and motivation. They are detailed as below.

1. Motivation. It is not clear why we need such attack. Existing studies already show that mainstream T2I models have the bias issue, and it is very easy to generate biased content. With some techniques, like jailbreak, this task will be easier. In this case, why should we still need this method? In the evaluation part, I could not find baseline comparisons over existing solutions for biased content generation.

2. I have quite a lot of doubts about the threat model. I found a lot of conflicting sentences in this paper. First,  who will activate the backdoor for biased content generation? According to Line 39, "a benign user includes these triggers in their prompts...". Then the question is: how does the benign user knows the trigger? Since he is benign, he will not intentionally include the trigger. He must do this accidently. Then, what is the chance of including the trigger? According to Line 149, "Our attack is both difficult to detect....". This indicates that it is very hard to trigger the backdoor accidently due to the combination of two triggers. Then this will be conflicting with the benign user statement. If the benign user could accidently activate the backdoor, then it will be easy to detect the backdoor, since it is easy to activate.

3. Second, in Section 3.2, the authors mentioned one scenario is that Company/API as Adversary. I am not sure how such adversary is able to poison the training/fine-tuning data. Another scenario is the Open-Source Platform Exploitation, where the adversary embeds a backdoor into his own model and releases it to the public. This does not make sense that an adversary attacks his own model. This will compromise his reputation, and is easily be tracked to his identify.

4. Again, in Section 3.2, the authors claim that the adversary can inject some samples into the training/fine-tuning data. It seems the evaluation only considers the fine-tuning scenario? Do the authors evaluate the case of training the foundation model? I am not sure as the experiment configuration details are missing. If not, then the evaluation is different from the claimed threat model section.

5. Technically, the proposed method has no novelty. It is simply a backdoor attack against T2I model. Selecting two words as the trigger is not novel.

**Questions:**

1. Could you clearly describe your threat model, e.g., who is the attacker, and who is the victim? what is the motivation of the attacker?

2. Highlight your technical contributions of the solution.

---

> ### Author Response · Authors · 2024-11-21
>
> We thank the reviewers for their valuable feedback.
>
> **1 & 5**. The reviewer asked about the motivation and novelty of this paper.
>
> We have introduced various novel aspects that distinguish it from prior works:
>
> **Accessibility and Cost-Efficiency**: We demonstrate a pipeline that enables anyone with query access to generative models to launch backdoor attacks by generating poisoning samples easily and at a very low cost. This broadens the potential range of attackers beyond those with advanced technical skills or resources.
>
> **Realistic Objectives**: Unlike previous works (e.g., [1]), which focus on objectives like generating outputs entirely unrelated to the user’s prompt, we focus on bias injection. Such objectives in earlier works are often easily detectable with a single generation, enabling the target model to address the issue swiftly. In contrast, bias injection is more subtle, can influence generations over the long term, and requires advanced techniques and multiple evaluations to detect.
>
> **Triggers Selection**: Many earlier works rely on unnatural or unrealistic triggers, such as special characters ([1]), which are rarely used by real-world users. Our work targets *natural language triggers*, making the attack both practical and harder to detect.
>
> **Data Availability**: One significant challenge in previous poisoning attacks was the lack of suitable data for *arbitrary goals*. Our work leverages current generative models to fill this gap, showing that it is now feasible to generate the necessary data efficiently for any arbitrary goal.
>
> **Stealth and Detection Resistance**: Our method improves upon earlier approaches that often produced easily detectable poisoned samples, such as those with a mismatch between caption and image. Instead, our pipeline produces highly aligned pairs of captions and images, greatly complicating the task of detection and filtering by APIs.
>
>
> **Threat Model and Comparison with Jailbreaking**
> The threat model in our approach differs significantly from jailbreak attacks. In jailbreak attacks, the adversary is a malicious user attempting to craft a prompt that leads to harmful or otherwise undesired output. However, in our work, the adversary could be anyone with access to state-of-the-art generative models for generating poisoning content, and the victim is any benign user of these text-to-image models. Additionally, it is unclear how jailbreaking could make this task easier. For instance, if the adversary targets the words "doctor" and "reading" as triggers and aims to associate them with the bias "being dark-skinned," how could jailbreaking achieve this specific attack? Even if it were feasible, there is no clear victim in such a scenario.
>
> **2**. The reviewer raised concerns about the threat model. Here, the adversary is anyone with access to state-of-the-art generative models, and the victim is any benign user of these APIs. Our pipeline balances bias exposure and detectability by selecting triggers: frequent combinations for visibility or rare ones for stealth. Even if a backdoor is activated, detecting it is difficult with a few examples. As noted, the backdoored model maintains high text-image alignment—e.g., a prompt for "a boy eating food" would display that, with a hidden bias like a Nike logo subtly included. In contrast, prior attacks aimed for obvious manipulations (e.g., a dog instead of a cat), making detection easier.
>
> **3**. The reviewer raised concerns about two threat model scenarios. In **Open-Source Platform Exploitation**, an adversary uploads a backdoored model to platforms like Hugging Face, using deceptive names or URLs. This aligns with previous work [1] involving domain spoofing or malicious service providers. Such adversaries prioritize spreading backdoors over maintaining reputation.
>
> In the **Company/API as Adversary** scenario, a company could subtly inject biases or propaganda to influence societal attitudes, possibly offering free services to undermine competitors. This scenario underscores the diverse and complex nature of potential threats.
>
> **4**. The reviewer asked about evaluating both training and fine-tuning scenarios. There was a typo in the paper: "pre-training" should be "continual training." While models may undergo pre-training, continual training, or fine-tuning, we focused on a large-scale continual training/fine-tuning experiment with 200,000 samples, injecting 400 poisoning samples to assess the attack's impact. This demonstrates the method's feasibility in realistic settings. We'll clarify this in the revised version.
>
> **Q1**: For the technical contributions, please refer to our earlier response (1 & 2) where we provided more details on the threat model.
>
> **Q2**: For the technical contributions, please refer to our earlier responses (1) where we outlined the novelty of our approach.
>
> [1] Struppek et al., Rickrolling the Artist: Injecting Backdoors into Text Encoders for Text-to-Image Synthesis, ICCV 2023

---

> > ### Comment · Reviewer_tMot · 2024-11-23
> >
> > Thanks for the response. However, I am still not convinced by the threat model.
> >
> > It is stated that the benign users are the victim. It requires the victim to inject the trigger to be attacked. This is very counterintuitive. In a backdoor, it is usually the attacker that injects the trigger. I never saw an attack that requires the victim to inject the trigger. This will bring several questions.
> >
> > 1. How could the victim inject the trigger? This must be done accidently. Then what is the probability of accidently injecting such trigger?
> >
> > 2. The attack is totally uncontrollable. The attacker cannot control which user could be attacked, and when he/she will be attacked. It all depends on the probability of injecting the trigger.
> >
> > 3. If the probability is low, then the attack will be very weak. If the probability is high, it means a lot of users will receive biased content. In this case, they will report to the model service provider, and the provider will detect that the model is under attack. Besides, if the probability is high, it is also very easy for the model owner to test and find the attack.
> >
> > In summary, I still find the threat model is not reasonable, and has some conflicts in practical scenarios.

---

### Note · Authors · 2025-01-14

I have read and agree with the venue's withdrawal policy on behalf of myself and my co-authors.